# Real-Time fMRI Neurofeedback Training of Selective Attention in Older Adults

**DOI:** 10.3390/brainsci14090931

**Published:** 2024-09-18

**Authors:** Tian Lin, Mohit Rana, Peiwei Liu, Rebecca Polk, Amber Heemskerk, Steven M. Weisberg, Dawn Bowers, Ranganatha Sitaram, Natalie C. Ebner

**Affiliations:** 1Department of Psychology, University of Florida, Gainesville, FL 32611, USA; peiweiliu@berkeley.edu (P.L.); r.polk@ufl.edu (R.P.); aheemske@asu.edu (A.H.); smweis@gmail.com (S.M.W.); natalie.ebner@ufl.edu (N.C.E.); 2Institute of Biological and Medical Engineering, Department of Psychiatry and Section of Neuroscience, Pontificia Universidad Católica de Chile, Santiago 7820436, Chile; neuro.mrana@gmail.com; 3Department of Clinical and Health Psychology, University of Florida, Gainesville, FL 32610, USA; dawnbowers@phhp.ufl.edu; 4St. Jude Children’s Research Hospital, Memphis, TN 38105, USA; ranganatha.sitaram@stjude.org; 5Center for Cognitive Aging and Memory, McKnight Brain Institute, University of Florida, Gainesville, FL 32610, USA

**Keywords:** rtfMRI, neurofeedback training, old age, selective attention, dACC

## Abstract

Background: Selective attention declines with age, due to age-related functional changes in dorsal anterior cingulate cortex (dACC). Real-time functional magnetic resonance imaging (rtfMRI) neurofeedback has been used in young adults to train volitional control of brain activity, including in dACC. Methods: For the first time, this study used rtfMRI neurofeedback to train 19 young and 27 older adults in volitional up- or down-regulation of bilateral dACC during a selective attention task. Results: Older participants in the up-regulation condition (experimental group) showed greater reward points and dACC BOLD signal across training sessions, reflective of neurofeedback training success; and faster reaction time and better response accuracy, suggesting behavioral benefits on selective attention. These effects were not observed for older participants in the down-regulation condition (inverse condition control group), supporting specificity of volitional dACC up-regulation training in older adults. These effects were, unexpectedly, also not observed for young participants in the up-regulation condition (age control group), perhaps due to a lack of motivation to continue the training. Conclusions: These findings provide promising first evidence of functional plasticity in dACC in late life via rtfMRI neurofeedback up-regulation training, enhancing selective attention, and demonstrate proof of concept of rtfMRI neurofeedback training in cognitive aging.

## 1. Introduction

It is crucial that individuals allocate their limited attention selectively to relevant information (i.e., targets) and inhibit irrelevant information (i.e., distractors). For example, when having a conversation in a crowded space, we pay attention to what our conversation partner says while blocking out background noise [1]. The capacity for selective attention, however, declines with age [2,3]. This age-related decline has been attributed to functional changes in the dorsal anterior cingulate cortex (dACC) [4,5]. Specifically, neuroimaging evidence supports dACC involvement in selective attention tasks, e.g., the Stroop Task [6], the Eriksen Flanker Task [7], and the Multi-Source Interference Task (MSIT) [8], including in older adults [9]. These tasks require cognitive processing such as target detection, novelty/salience indication, and conflict monitoring [10,11]. In addition, dACC is involved in higher-order complex processes pertaining to executive function, decision making, and cognitive control [12,13,14,15], and in these roles it has been shown to be functionally connected to other brain regions, such as the insula, amygdala, orbital frontal cortex, medial prefrontal cortex, superior temporal cortex, and thalamus [16], and serve as a key node in large-scale brain networks (e.g., the salience network and the default mode network) [17,18].

Cognitive intervention studies have demonstrated the trainability of selective attention in older adults [19]. For example, one experiment used the MSIT, which produces robust interference effects by asking participants to indicate as quickly and accurately as possible which of the three digits is different from the other two (e.g., 100, 232), regardless of its position [20], and found that older adults who received multi-task training were more accurate on the task than a control group that received no training [21]. Critically, training-induced behavioral improvement was accompanied by greater dACC activity, suggesting a possible neural mechanism underlying the behavioral training effects. Building on this evidence, the present study examined the extent to which real-time functional magnetic resonance imaging (rtfMRI) neurofeedback could be used to train older adults to up-regulate dACC activity (i.e., neurofeedback training success) with associated improvements on selective attention (i.e., behavioral benefit). 

rtfMRI neurofeedback training is an advanced neuroimaging technique in which participants are instructed to attempt regulation of their brain activity based on feedback. The feedback can be visually presented and reflects a near-real-time blood oxygen level-dependent (BOLD) signal in a region (or regions, or networks) of interest [22,23] (see also [24,25] for diverse basic science and clinical applications of rtfMRI neurofeedback training protocols and a discussion of theories underlying this technique). rtfMRI neurofeedback training has been successfully applied in healthy young adults [26,27,28,29] as well as young/middle-aged adult clinical populations (e.g., schizophrenia, mean age 26.3 years [30]; attention deficit hyperactivity disorder (ADHD), mean age 36.9 years [31]). However, the impact of rtfMRI neurofeedback training on enhancing behavior is still controversial [23]. For example, with specific relevance to rtfMRI neurofeedback training of dACC activity, Zilverstand and colleagues found that individuals with ADHD learned volitional dACC up-regulation with subsequent behavioral benefits on selective attention, working memory, and inhibition [31]. 

While some studies have examined the effectiveness of rtfMRI neurofeedback training in older adults with neurodegenerative disorders (e.g., Alzheimer’s disease (AD)), with mixed results [32,33,34], the use of this technique in older adults is still limited [24]. As one exception, Hohenfeld and colleagues trained 16 healthy older adults (63.5 ± 6.7 years) and older adults with AD (66.2 ± 8.9 years) to up-regulate parahippocampal gyrus activity via rtfMRI neurofeedback [32]. However, while all participants in the experimental condition (compared to four healthy older adults (64.8 ± 9.5 years) in the sham-feedback control condition) showed behavioral improvement in visuospatial memory performance from pre- to post-training, this behavioral effect was not accompanied by increased activity in parahippocampal gyrus. A similar study by the same group reported no changes either in behavior nor in associated neural activity from pre- to post-training for adults in the contingent neurofeedback training condition [33]. 

To advance knowledge regarding dACC functional plasticity and trainability of selective attention in older adults, the present study had two specific aims. The first aim was to test the aging brain’s capacity to learn volitional up-regulation of dACC activity via rtfMRI neurofeedback training in the context of a selective attention task (*Neurofeedback Training Success; Aim 1*). In particular, we hypothesized that both young and older adults would show neurofeedback training success using rtfMRI-guided training in up-regulation of dACC activity. Based on robust evidence of functional decline in dACC activity with age [35,36] and reduced functional neuroplasticity in aging [37], we expected this effect, while present, to be less pronounced in older adults (our experimental group; i.e., older adults receiving dACC up-regulation training) than young adults (our age control group; i.e., young adults receiving dACC up-regulation training). 

The second aim was to examine subsequent effects of neurofeedback training on enhancing selective attention in older age (*Behavioral Benefits from Neurofeedback Training; Aim 2*). In particular, we hypothesized a behavioral benefit from the rtfMRI neurofeedback training on selective attention to be present only for dACC up-regulation (experimental group) but not dACC down-regulation (inverse condition control group; i.e., older adults receiving dACC down-regulation training), given evidence of enhanced dACC activity in greater selective attention [10,11]. 

To this end, going conceptually and methodologically beyond previous work, we used rtfMRI neurofeedback to train a group of older adults (covering a wider and older age range than prior studies; but see [32]) in volitional up-regulation of bilateral dACC activity (experimental group) during the MSIT and compared their behavioral changes in task performance over the course of the neurofeedback training. We recruited two control groups: a group of young participants (i.e., age control group) who underwent the same up-regulation rtfMRI neurofeedback training protocol as the experimental group, which allowed us to directly test age-group differences in neurofeedback training success; and a group of older participants who underwent a bilateral dACC down-regulation rtfMRI neurofeedback training protocol (i.e., inverse condition control condition) [23,38]) to speak to the specificity of our dACC up-regulation training effects on selective attention. To complete the 2 × 2 design, we furthermore included a group of young adults who underwent bilateral dACC down-regulation neurofeedback training. 

## 2. Methods

### 2.1. Participants 

A total of 46 right-handed participants with no major psychiatric disorders were recruited for this study, including 27 older adults (*M*age = 70.04 years, *SD* = 7.37 years, range: 57–86 years, 44.4% female; According to the National Institutes of Health, older adults are defined as 65 years and older (https://www.nih.gov/nih-style-guide/age#older-adults; accessed on 2 September 2024). In our sample, age ranged from 57–86, with 8 participants being younger than 65. However, for better readability, we refer to our sample of middle-aged and older adults as “older adults” throughout the paper) and 19 young adults (*M*age = 20.76, SD = 1.44 years, range: 19–23 years, 52.6% female). Participants were recruited through Institutional Review Board (IRB)-approved registries, university clinics, the HealthStreet community outreach program, and other community recruitment efforts (flyers, handouts, word of mouth). All participants underwent a phone prescreening (~30 min) to determine study and MRI eligibility, and older participants completed the Telephone Interview for Cognitive Status (TICS; passing score ≥ 30) [39] and the Montreal Cognitive Assessment (MoCA; passing score ≥ 22) [40] for cognitive screening. 

Leveraging methodological approaches in clinical randomized controlled trials [41], we employed unequal randomization, resulting in unbalanced group sizes by over-sampling the experimental group (i.e., older adults in the up-regulation condition) at a 2:1 ratio compared to the other/control groups to (i) enhance statistical power for detecting within-subject/participant-level neurofeedback training effects in this primary group of interest; (ii) increase the amount of information on the new neurofeedback training approach in this group; while (iii) allowing for considerable cost savings As a result, 18 older participants (experimental group; 50.0% female) and 10 young participants (age control group; 38.9% female) were randomized into the up-regulation condition (five participants in the experimental group were diagnosed with Parkinson’s disease (PD), and 4 reported subjective cognitive decline (SCD; i.e., indicated that they felt like their memory was becoming worse; that this concerned them; and that they had a family history of AD or related dementias). Note, however, that performance on the TICS [39], the MoCA [40], and four cognitive tasks from the NIH Cognition Toolbox (i.e., Flanker Inhibitory Control and Attention Test, Dimensional Change Card Sort Test, List Sorting Working Memory Test, and Pattern Comparison Processing Speed Test) [42] did not differ in these 9 individuals from performance of the rest of the older participants in this study. In addition, the reported effects did not change when PD and SCD status was considered as covariate in the analyses. Also, see Appendix A for comparability of reaction time and response accuracy in the MSIT among participants with PD/SCD and the remaining older adults); 9 older participants (inverse condition control group; 55.6% female) and 9 young participants (young down-regulation group; 55.6% female) were randomized into the down-regulation condition. 

Table 1 provides sample-descriptive information pertaining to cognition, affect, and health at pre-training in each of the four groups. As shown, the experimental group performed worse than the age control group on three, worse than the inverse condition control group on one, and worse than the young down-regulation group on all four cognitive tasks from the NIH Cognition Toolbox [42]. The experimental group reported higher positive affect and lower negative affect than the young down-regulation group, while they reported comparable positive and negative affect with the age control and the inverse condition control groups. Finally, the experimental group reported comparable physical/mental health with the other three groups.

### 2.2. Procedure 

The study was approved by the IRB at the University of Florida (#201300801) in July 2014. All participants provided informed consent before enrollment. The study comprised a pre-training session, followed by seven neurofeedback training sessions, scheduled one to two days apart. Each participant completed the study within three weeks of enrollment. 

#### 2.2.1. Pre-Training Session

In the pre-training session, study consent was obtained, and participation eligibility was confirmed. For sample-descriptive purposes and to allow comparison to other samples in the cognitive aging literature, participants also completed four cognitive tasks from the NIH Toolbox Cognition Battery IPAD application, including the Flanker Inhibitory Control and Attention Test, Dimensional Change Card Sort Test, List Sorting Working Memory Test, and Pattern Comparison Processing Speed Test [42], responded to the Positive and Negative Affect Schedule [43], and responded to two single items measuring physical and mental health respectively. 

#### 2.2.2. Neurofeedback Training Sessions

Procedures were identical across the four groups, unless noted otherwise. The neurofeedback training comprised of seven sessions (Figure 1A). In the first neurofeedback training session, participants confirmed their MRI eligibility, then received instructions for the MSIT and completed a short practice (see below for details for the MSIT) before entering the scanner. During the practice, participants were instructed to complete the MSIT trials as quickly and accurately as possible. They were also informed that the reward points later on, when they performed the task in the MRI scanner, would be determined by their ability to regulate their brain activity and that their goal was to gain as many points as possible on each trial, with total points earned translating into a monetary bonus payout of up to $100 at the end of the study.

Next, participants were placed in the MRI scanner. We acquired a 3 min T1-weighted (T1) image, followed by an 8 min resting-state functional scan (not analyzed here). Participants then completed two runs of a functional localizer task (see details in the Appendix A), followed by a magnetic resonance spectroscopy (MRS) scan (not analyzed here). During the MRS scan, data from the localizer task were analyzed to compute individualized region of interest (ROI) masks for bilateral dACC for each participant (see Figure 2A). Finally, participants completed the first three neurofeedback training runs (see details in *Neurofeedback Training Task* below). 

The second through sixth neurofeedback training sessions were all identical. Each session occurred on a separate day and began with confirmation of MRI eligibility and short reminders of the instructions for the MSIT and the neurofeedback training runs. Inside the scanner, after a T1 image, participants completed six neurofeedback training runs per session. In the last (seventh) neurofeedback training session, again after MRI eligibility confirmation that day, participants completed a T1 image, three neurofeedback training runs, an 8 min resting-state functional scan, and an MRS scan.

Outside the scanner, each training session concluded with a short post-event questionnaire inquiring about the participant’s experience during the MRI, as well as a detailed report about total points earned during that day’s neurofeedback training runs. Participants received a $300 fixed compensation for completion of the study and a bonus reward, which was based on the points earned during the neurofeedback training runs (1 point = $0.008; $100 maximum). 

### 2.3. Image Acquisition and Online Data Preprocessing

Brain images were collected on a 3T Siemens Prisma scanner (Siemens Medical Solutions USA: Malvern, PA, USA) with a 64-channel head coil. Only imaging parameters of scans relevant to the analysis here are reported. Whole-brain high-resolution three-dimensional T1 anatomical reference images were acquired using a magnetization-prepared rapid acquisition gradient-echo (MP-RAGE) sequence [46] (208 sagittal planes, field of view (FOV) = 256 × 256, voxel size (VS) = 1 mm^3^, repetition time (TR) = 2000 ms, echo time (TE) = 2.91 ms, inversion time (TI) = 1010 ms, flip angle (FA) = 8°), generalized auto-calibrating partially parallel acquisition (GRAPPA) [47] acceleration factor = 2, with 24 reference lines. All functional images (localizer, neurofeedback training) were acquired with a multi-band echo-planar imaging (EPI) sequence (50 slices along the AC-PC line, FOV = 240 × 240, TR = 1500 ms, TE = 30 ms, VS = 3 mm^3^ without gap, FA = 70°, phase-encoding (PE) direction acceleration factor = 2, slice direction acceleration factor = 2, bandwidth = 2232 Hz/Px). 

All online analyses during data acquisition were conducted on preprocessed imaging data. Preprocessing used custom MATLAB R2014a code based on Statistical Parametric Mapping (SPM) functions [48]. Structural image preprocessing used the T1 image from the first neurofeedback training session and included dicom-nifti conversion, segmentation, and normalization. The processing of functional images from the localizer and the neurofeedback training runs included dicom-nifti converting, realignment, co-registration to the preprocessed T1 image, and normalization.

### 2.4. Neurofeedback Training Task

As shown in Figure 1B, each neurofeedback training run consisted of four blocks, which each consisted of 12 MSIT trials: one baseline block and three regulation blocks (up- or down-regulation, depending on the group participants were in). The baseline block served to determine levels of BOLD signal for each run in bilateral dACC and primary auditory cortex (PAC; a control region that was selected as it was unrelated to the specific task demands here [49,50]) needed for calculation of the neurofeedback (see details below). The inter-block interval was 7.5 s, during which a fixation cross was presented. 

As shown in Figure 1C, each MSIT trial during the neurofeedback training runs started with a fixation cross for 1.5 s, followed by task stimuli presentation for 1 s and then a black screen for 3.5 s. At the end of each trial, for regulation blocks, the neurofeedback in the form of reward points appeared on the screen for 1.5 s; for baseline blocks, no neurofeedback/reward points were presented (i.e., “* Points” appeared on the screen); and for trials in which participants missed to give a response, a “No Response” message appeared.

On each MSIT trial, a three-digit number (e.g., 212) appeared on the screen (Figure 1C). Participants were instructed to indicate the unique digit (which could be 1, 2, or 3) via button press with their index, middle, and ring finger of their right hand, respectively. The three-digit numbers used here were identical to those used in interference trials in Bush and Shin [20]. In each trial the digits were overlaid over a face (with a happy or angry expression, or a scrambled image, see https://faces.mpdl.mpg.de/imeji/ for details; accessed on 2 September 2024), as an additional source of interference; and participants were instructed to attend to the numbers while ignoring the face in the background. We selected happy and angry facial expressions to align with Ebner and Johnson [51], who previously used this task in young and older adults to determine age-related differences in multi-source interference/selective attention. The MSIT produces two behavioral outcome measures for each neurofeedback training run: reaction time, which was the average time across all correct trials during regulation blocks; and response accuracy, which was the total count of correct trials during regulation blocks (theoretical range: 0–36). 

Identical for both the up- and down-regulation training conditions, participants were asked to work as quickly and accurately as possible and to stay focused on the numbers in each trial. Participants were also informed that after the initial trials for each run, trials would award points based on their ability to regulate their brain activity and that the goal was to obtain as many points as possible. Each run lasted ~7.5 min, and at the end of each run, participants were reminded that the goal was to gain as many points as possible on each trial and to try to increase their points as much as possible. Immediately after each run, participants reported their motivation to continue the task (e.g., “On a scale from 1 to 100, how motivated are you to continue the task?”), which gave us a numeric indicator of participants’ self-reported level of motivation to continue with the task after each run. 

### 2.5. Online Data Analysis and Neurofeedback/Reward Points Calculation

Neurofeedback during the training task was calculated based on BOLD signal during the 3 s data collection period (2 TRs; the time window covered by the gray frame in Figure 1C) and presented in the form of reward points. Reward points were calculated in two steps. First, we calculated the BOLD signal difference between bilateral dACC (target ROI) and PAC (control ROI) for each trial via the following Equation (1).
(1)$${\mathrm{BOLD}}_{\mathrm{diff}}=\frac{1}{2}\sum_{t=1}^{2}\left({BOLD}_{ROI\;t}\left[\begin{array}{c} 1\\ 1\\ -1\\ -1 \end{array}\right]\right)$$

BOLD*_ROI t_* is a (4 × 1) matrix that contains BOLD signal from left dACC, right dACC, left PAC, and right PAC in a given TR *t*. The component in the parentheses refers to the difference of BOLD signal across bilateral dACC in contrast to bilateral PAC. This difference score was averaged across 2 TRs within a given trial.

We then calculated the reward points for a given trial during regulation blocks using Equation (2):(2)$${RewardPoints}_{\left(ct\right)}=\frac{c\cdot ({BOLD}_{diff\left(ct\right)}-mean\left({BOLD}_{diff\left(bl\right)}\right))}{std({BOLD}_{diff\left(bl\right)})}$$

In this equation, c represents the condition factor (1 = up-regulation, −1 = down-regulation). Subscripts ct and bl refer to the current trial and baseline trial, respectively. BOLD*_diff_*_(ct)_ refers to the BOLD signal difference between bilateral dACC and PAC in the current trial. The mean(BOLD*_diff_*_(bl)_) and std(BOLD*_diff_*_(bl)_) refer to the mean and standard deviation of the BOLD signal difference between bilateral dACC and PAC across trials in the baseline block. 

Note that calculation of the neurofeedback was identical for the up- and the down-regulation condition; other than that, the neurofeedback (in the form of reward points) for the up-regulation protocol (i.e., experimental group, age control group) was based on an increase in dACC BOLD signal (relative to BOLD signal in PAC) in regulation relative to baseline blocks; while neurofeedback (in the form of reward points) for the down-regulation protocol (i.e., inverse condition control group, young down-regulation group) was based on a decrease in dACC BOLD signal (relative to BOLD signal in PAC) in regulation relative to baseline blocks. 

The numerical outcome of Equation (2) was rounded to two decimals and then presented to participants as neurofeedback in the form of points (Figure 1C). The theoretical numerical outcome from Equation (2) could be any value from negative infinity to positive infinity, but when the outcome was lower than zero, points were displayed as zero; when the outcome was higher than ten, points were displayed as ten. This procedure allowed to limit the range of the reward points given out and facilitated conversion into the monetary bonus paid out at the end of the training protocol. 

### 2.6. Analysis on fMRI Data to Confirm dACC Involvement in Localizer Task and Neurofeedback Training

#### 2.6.1. Localizer

Data from the localizer runs were preprocessed using the standardized fMRIPrep pipeline (see Appendix A for details). We then fitted a GLM using SPM12 (https://www.fil.ion.ucl.ac.uk/spm/software/spm12/; accessed on 2 September 2024). For each participant, the first-level model included eight regressors (e.g., four regressors that corresponded to the time series of one of each of the four MSIT trial types; see Appendix A) plus six regressors corresponding to the head motion reflective of six dimensions. All regressors were convolved with the canonical hemodynamic response function (HRF). Within each first-level model, a *t*-contrast was computed to identify the brain regions with greater activity during the most difficult MSIT trial type (which presented interference digits with faces in the background) compared to the least difficult MSIT trial type (which presented control digits without faces in the background). The parameter estimate of this *t*-contrast for each participant was used to create a group-level random effect model, with age group (young vs. older) and training condition (up- vs. down-regulation) as two between-subject factors. 

We then conducted a one-sample *t*-test across all participants to confirm involvement of dACC in the MSIT and verify the localization of the dACC ROI in the online analysis. This one-sample *t*-test also identified additional brain regions recruited during the localizer runs (see Appendix A for a comprehensive list and visualization).

Finally, we conducted additional contrasts on the second (group-average) level for the localizer data to confirm that there were no pre-training differences in dACC activity during the MSIT between our four groups in support of successful randomization: the t-contrast young vs. older, the *t*-contrast up- vs. down-regulation, and the F-contrast: age × training condition. Clusters that consisted of more than 10 voxels with an FWE-corrected *p*-value smaller than 0.05 were considered statistically significant.

#### 2.6.2. Neurofeedback Training

We used the same fMRIPrep preprocessing pipeline for the brain data collected during the neurofeedback training runs. Then, we constructed a GLM using SPM 12 (https://www.fil.ion.ucl.ac.uk/spm/software/spm12/; accessed on 2 September 2024) for each training run, in which three regressors of interests (e.g., time series of fixation, stimuli presentation, and reward presentation) plus six head motion regressors were defined. All regressors were convolved with the canonical hemodynamic response function (HRF). To determine brain activation associated with the MSIT, we conducted a t-contrast to identify regions with greater activity for stimuli (digits/faces) than fixation for each training run. The parameter estimate of this t-contrast for each training run within the same participant was used to create a participant-level random effect model. Then, parameter estimates of this t-contrast across all runs for each participant were extracted to create a group-level random effect model, with age (young vs. older) and training condition (up- vs. down-regulation) as two between-subject factors.

We again conducted a one-sample *t*-test across all participants to confirm the involvement of dACC during the neurofeedback training and verify the localization of the dACC ROI in the online analysis. This one-sample *t*-test also identified additional brain regions recruited during the neurofeedback training (see Appendix A for a comprehensive list and visualization). Clusters that consisted of more than 10 voxels with an FWE-corrected *p*-value smaller than 0.05 were considered statistically significant.

### 2.7. Analysis of Neurofeedback Training Effects

#### 2.7.1. Neurofeedback Training Success (Aim 1)

We used *(i) reward points* and *(ii) averaged dACC BOLD signal* during the MSIT as the two outcome variables reflective of neurofeedback training success (*Aim 1*). In all groups, reward points were computed as the sum of points gained in each neurofeedback training run (see details above). The dACC BOLD signal was extracted from the dACC ROIs identified via the individualized localizer and averaged across the two TRs per trial that were used for reward point calculation (Figure 1C). To control for baseline differences in BOLD signal between runs, we used the ratio of dACC BOLD signal to global BOLD signal.

#### 2.7.2. Behavioral Benefits from Neurofeedback Training (Aim 2)

Furthermore, we used *(iii) reaction time* and *(iv) response accuracy* in the MSIT as the two outcome variables reflective of behavioral benefit from neurofeedback training (*Aim 2*).

To accommodate for the nested data structure (i.e., sessions and runs nested under participants), we used multilevel regression. For each of the outcome variables (reward points, dACC BOLD signal, reaction time, response accuracy), a separate model was created. In each model, session, run, and their interaction served as predictors, with run centered within session and session centered at the grand mean. To further determine the specificity of the training effects, we examined the moderation of group (experimental, age control group, inversion condition control group, and young down-regulation group) on the effects of session and/or run for each of the four outcome variables. In this analysis, the experimental group constituted the reference group, and session and/or run model estimates indicated the effects in this group. Interaction effects indicated the extent to which the other three groups differed from the experimental group. We followed up on significant moderations with separate within-group analyses. All multilevel regression analyses were conducted using Stata16.1 [52].

## 3. Results

### 3.1. Confirmation of dACC Involvement in the Localizer Task and the Neurofeedback Training

One-sample *t*-tests across all participants in the second-level model for both the localizer (Figure 2B) and the neurofeedback training (Figure 2C) runs identified clusters that overlapped with peak voxels in bilateral dACC previously reported for the MSIT (left dACC MNI: −10, 10, 49; right dACC MNI: 3, 10, 47 [8]; see also Appendix A). These dACC clusters were also well aligned with the individualized ROI masks for bilateral dACC that we identified for each participant in our localizer task during the online analysis (Figure 2A). These findings indicated good localization of dACC as the target ROI in our neurofeedback training protocol.

No significant clusters at *p* < 0.05 (FWE corrected) were identified for the t-contrast young vs. older, the t-contrast up- vs. down-regulation, or the *F*-contrast age × training condition at pre-training, supporting successful randomization into our four groups. 

### 3.2. Neurofeedback Training Effects

#### 3.2.1. Neurofeedback Training Success (Aim 1)

Across the whole training protocol, *reward points* received by participants in the experimental group were significantly lower than those received by participants in the age control group (*B* = 17.61, *z* = 2.14, *p* = 0.032) but were higher than those received by participants in the inverse condition control group (*B* = −17.27, *z* = −2.03, *p* = 0.042). Reward points did not significantly differ between the experimental group and the young down-regulation group (*B* = −3.62, *z* = −0.43, *p* = 0.67). Confirming our prediction under *Aim 1*, the effect of session on reward points was significant for the experimental group (*B* = 3.58, *z* = 3.12, *p* = 0.002), indicating that older participants in the up-regulation condition were able to increase their reward points over the course of the training sessions (Figure 3A). This session effect on reward points in the experimental group significantly differed from the session effect in the other three groups (vs. age control group: *B* = −4.98, *z* = −2.61, *p* = 0.009; vs. inverse condition control group: *B* = −7.33, *z* = −3.67, *p* < 0.001; vs. young down-regulation group: *B* = −5.43, *z* = −2.69, *p* = 0.007). In fact, this session effect was not significant in either the age control group (*B* = −1.39, *z* = −0.84, *p* = 0.40) nor the young down-regulation group (*B* = −1.86, *z* = −1.11, *p* = 0.27), and was significant, but in the negative direction, in the inverse condition control group (i.e., decreasing reward points across training sessions; *B* = −3.73, *z* = −2.62, *p* = 0.009). 

Furthermore, the effect of run was not significant in the experimental group (*B* = −0.53, *z* = −0.42, *p* = 0.68), but the run effect in the experimental group was significantly different from the run effect in the age control group (*B* = −9.34, *z* = −4.35, *p* < 0.001). Follow-up analysis showed a significant negative effect of run in the age control group (*B* = −9.87, *z* = −5.24, *p* < 0.001), in that reward points decreased over the course of the runs in young adults in the up-regulation condition. See Appendix A for visualization of run effects across the neurofeedback training sessions. No other effects on reward points were significant (all *p*s > 0.05).

Consistent with our findings for reward points, the session effect on *dACC BOLD signal* was significant for the experimental group (*B* = 0.001, *z* = 2.10, *p* = 0.035), indicating an increase in dACC BOLD signal in older participants in the up-regulation condition across the neurofeedback training sessions, in line with our prediction under *Aim 1* (Figure 2B). Again, the session effect for the dACC BOLD signal in the experimental group significantly differed from the session effects in the age control group (*B* = −0.002, *z* = −2.29, *p* = 0.022) as well as the inverse condition control group (*B* = −0.03, *z* = −3.24, *p* = 0.001), but was not significantly different from the session effect in the young down-regulation group (*B* = −0.0003, *z* = −0.31, *p* = 0.75). Further, follow-up analyses showed that the effect of session on dACC BOLD signal was not significant for the age control group (*B* = −0.001, *z* = −1.16, *p* = 0.25; or the young down-regulation group (*B* = 0.001, *z* = 1.06, *p* = 0.29), while the session effect was significant, but again in the negative direction, for the inverse condition control group (*B* = −0.002, *z* = −2.42, *p* = 0.016). However, this negative linear effect of session on dACC BOLD signal in the inverse condition control group did not align with the negative linear effect of session on reward points, suggesting that rtfMRI neurofeedback-driven learning may not have been at work. In fact, the dACC BOLD signal showed a significant negative session effect for participants in the inverse condition control group during baseline trials (χ^2^(1) = 4.19, *p* = 0.04). Thus, given the present study’s calculation of the neurofeedback, a possible explanation for the pattern of findings in this control group is that the decrease in dACC BOLD signal across baseline trials resulted in no increase in reward points despite decreasing dACC BOLD signal across regulation trials. No other effects on the dACC BOLD signal were significant (all *p*s > 0.05).

#### 3.2.2. Behavioral Benefits from Neurofeedback Training (Aim 2)

Regarding *reaction time*, participants in the experimental group responded overall slower than participants in the age control group (*B* = −0.25, *z* = −4.20, *p* < 0.001) and in the young down-regulation group (*B* = −0.17, *z* = −2.84, *p* = 0.005), but not different from participants in the inverse condition control group (*B* = −0.09, *z* = −1.51, *p* = 0.13), in line with research on general age decline in processing speed [53]. Furthermore, the experimental group showed faster reaction time over the course of the sessions (*B* = −0.01, *z* = 3.15, *p* = 0.002), confirming our prediction under *Aim 2* (Figure 2C). This session effect in the experimental group was significantly different from the session effect in the age control group (*B* = 0.01, *z* = 2.15, *p* = 0.03), for which this session effect was not significant (*B* = 0.001, *z* = 0.51, *p* = 0.61). Also, the session effect for the experimental group was not significantly different from that for the inverse condition control group (*B* = −0.01, *z* = −1.95, *p* = 0.051) nor the young down-regulation group (*B* = 0.0002, *z* = 0.03, *p* = 0.98), and participants in these two control groups, just like those in the experimental group, showed faster reaction time over the course of the neurofeedback training sessions (inverse condition control group: *B* = −0.02, *z* = −4.65, *p* < 0.001; young down-regulation group: *B* = −0.01, *z* = −2.66, *p* = 0.008). No other effects on reaction time were significant (all *p*s > 0.05). 

Aligning with our findings for reaction time, we observed significant effects of both session (*B* = 0.62, *z* = 4.68, *p* < 0.001) and run (*B* = 0.29, *z* = 1.96, *p* = 0.05) on response accuracy for the experimental group, indicating that older participants in the up-regulation condition showed greater response accuracy across (as well as within) sessions, confirming our predictions under *Aim 2* (Figure 3D). These effects were further qualified by a significant session by run interaction (*B* = −0.21, *z* = −2.12, *p* = 0.03), in that increasing response accuracy across runs within a session reduced over the course of the training protocol. The session effect on response accuracy in the experimental group significantly differed from the session effect in the age control group (*B* = −0.55, *z* = −2.49, *p* = 0.013) and the inverse condition control group (*B* = −1.15, *z* = −4.99, *p* < 0.001) but not the young down-regulation group (*B* = 0.42, *z* = −1.83, *p* = 0.07). Follow-up analyses showed that the session effect on response accuracy was not significant for the age control group (*B* = 0.07, *z* = 1.58, *p* = 0.11) but was significant, in the negative direction, for the inverse condition control group (*B* = −0.53, *z* = −2.92, *p* = 0.003); that is, participants in the inverse condition control group showed worse response accuracy over the course of the neurofeedback training sessions. The interaction of session by run was also significantly different between the experimental group and the inverse condition control group (*B* = 0.42, *z* = −2.48, *p* = 0.013). In contrast to findings in the experimental group, neither the effect of run (*B* = −0.14, *z* = −0.71, *p* = 0.48) nor the session by run interaction (*B* = 0.21, *z* = 1.59, *p* = 0.11) were significant in the inverse condition control group, suggesting no linear change in response accuracy over the course of the runs within sessions in this control group. See Appendix A for visualization of run effects across the course of the neurofeedback training. No other effects on response accuracy were significant (all *ps* > 0.5). Given the wide age range among older adults in our sample (57–86 years), we conducted two control analyses on the older participants: (1) one in which we limited the sample to just older adults based on definitions by the NIH (65 years and older; *N* = 19); and (2) one in which we included chronological age (57–86 years) as a covariate. Results from these additional analyses closely aligned with those reported in the main text for all outcome variables.

#### 3.2.3. Follow-Up Brain-Behavior Analysis

We observed evidence for both neurofeedback training success and behavioral benefit in the experimental group. To test the link between neurofeedback training success and behavioral benefits, we conducted two multilevel regression models that examined effects of reward points and dACC BOLD signal, respectively, on reaction time and response accuracy in this group. In each model, reward points and the dACC BOLD signal of each neurofeedback training trial served as predictors. Reaction time and NMresponse accuracy for each trial served as outcome variables.

While the effect of reward points on reaction time was not significant (*B* = −0.001, *z* = −0.77, *p* = 0.44), more reward points were associated with better response accuracy in the MSIT (*B* = 0.002, *z* = 2.05, *p* = 0.04). Furthermore, greater dACC BOLD signal predicted faster reaction time (*B* = −0.26, *z* = −3.32, *p* = 0.001) and better response accuracy (*B* = 0.30, *z* = 3.00, *p* = 0.003), supporting a brain-behavior link in the experimental group. 

#### 3.2.4. Self-Reported Motivation (Post Hoc Analysis)

Our data support increased reward points and dACC BOLD signal over the course of the training sessions (neurofeedback training success) and shorter reaction time and greater response accuracy over the course of the training sessions (behavioral benefit from neurofeedback training) in the experimental group. However, neither neurofeedback training success nor behavioral benefit were observed in the age control group. Based on previous evidence demonstrating rtfMRI neurofeedback training success in dACC up-regulation for young adults with ADHD [31], the null effect observed in our age control group was rather surprising and not in line with our prediction. To explore a possible explanation for this unexpected finding, we conducted a multilevel regression model on participants’ self-reported motivation to continue the MSIT at the end of each run, with group (experimental vs. age control), session, and run, as well as their interactions, as independent variables. Neither the effects of session (*B* < −0.001, *z* = −0.0001, *p* = 0.999) and run (*B* = −0.50, *z* = −1.74, *p* = 0.08) nor their interaction (*B* = −0.11, *z* = −0.59, *p* = 0.55) were significant in the experimental group. That is, self-reported motivation in the experimental group did not change over the course of the neurofeedback training sessions/runs. However, the effect of run in the experimental group was significantly different from the run effect in the age control group (*B* = −4.61, *z* = −9.78, *p* < 0.001). Follow-up analysis within the age control group showed that, within a session, self-reported motivation decreased across runs (*B* = −5.12, *z* = −10.11, *p* < 0.001). Thus, one possible explanation for no neurofeedback training success in the age control group could be that they were insufficiently motivated to proceed with the MSIT/the neurofeedback training.

## 4. Discussion

This study examined for the first time the extent to which older adults can learn volitional control of dACC activity via rtfMRI neurofeedback training and determined the extent to which this training enhanced their selective attention. Our results showed a linear increase in the reward points as well as the BOLD signal from dACC along with training sessions in older participants from the up-regulation condition (e.g., the experimental condition). Furthermore, this group of participants also demonstrated improvement in the behavioral performance (e.g., shorter reaction time and increased response accuracy in the MSIT). Results provide crucial *proof of concept* of rtfMRI neurofeedback training in cognitive aging, supporting both neurofeedback training success (i.e., higher reward points and corresponding dACC BOLD signal over the course of the training sessions) and behavioral benefit (i.e., faster reaction time and better response accuracy in the MSIT) among older adults trained to up-regulate dACC activity in a sample with a wider and older age range than most prior work. We further demonstrate *specificity* of volitional dACC up-regulation training in older adults, as neither older adults in the down-regulation condition (inverse condition control group) nor—unexpectedly and possibly related to reduced motivation in this group to continue with the neurofeedback training pertaining to selective attention—young adults in the up-regulation condition (age control group) showed comparable effects. These findings provide promising first evidence of functional plasticity in dACC in old age via rtfMRI neurofeedback up-regulation training, which enhances selective attention, and, importantly, contribute to a currently still rather sparse literature on the feasibility and success of rtfMRI neurofeedback training in cognitive aging. 

Zilverstand and colleagues trained dACC up-regulation using rtfMRI neurofeedback in young and middle-aged adults with ADHD and demonstrated an increase in dACC activity from the second to the third (out of four) training session [31]. Going beyond their study and extending this finding to individuals in older age, the present study comprised 36 training runs across seven sessions. This longer training protocol allowed us to demonstrate change in the form of a linear increase in reward points and dACC BOLD signal across a larger number of sessions, that is, over a longer temporal scale, for demonstration of more sustainable neurofeedback training success.

Our instructions to participants entailed that the neurofeedback was based on their real-time brain activity. No explicit instructions were given as to what strategies should be used to drive this activity. Thus, neurofeedback training success observed in the experimental group likely reflects successful implicit learning [25,54] in these older adults. This finding is in line with the literature supporting intact implicit (relative to explicit) learning in aging [55,56]. It will be interesting to determine in future research the involvement of other brain regions than dACC in this learning process, such as the striatum and medial temporal regions [57,58].

A methodological challenge in rtfMRI neurofeedback training on cognitive function is to modulate (e.g., up- or down-regulate) activity in the target ROI and assure that target ROI activity reflects the behavior of interest. Previous studies often have instructed participants to engage in mental simulation during the regulation phase with the intention to trigger target ROI activity (e.g., remembering a real-world footpath to recruit parahippocampal gyrus [32]; mental calculation to recruit dACC [31]; and backward reciting digit/letter sequence to recruit dorsal lateral prefrontal cortex [29]). This methodology, however, could result in a dissociation between brain self-regulation via mental simulation and subsequently assessed behavior/cognition and may explain why a recent meta-analysis did not support behavioral benefit from rtMRI neurofeedback training [23]. Adopting a different approach, the present study asked participants to engage in the cognitive task/behavior of interest (the MSIT/selective attention) during the rtfMRI neurofeedback training, thus more directly associating in the study design dACC activity up-regulation (reflected both in increased reward points and greater dACC BOLD signal) with performance enhancement (i.e., shorter reaction time and greater response accuracy in the MSIT) over the course of the training sessions.

As noted, we observed evidence for both neurofeedback training success and behavioral benefit in the experimental group, in that older adults in the up-regulation condition demonstrated a linear increase in both reward points and dACC BOLD signal as well as faster reaction time and better response accuracy over the course of the training sessions. Our brain-behavior analysis, furthermore, showed that greater dACC activation during the neurofeedback training trials was directly correlated with faster and more accurate responding in the MSIT in the experimental group. This pattern of results renders simple practice effects as an explanation of the findings unlikely.

The inverse condition control group, in contrast, showed declines in reward points, dACC BOLD signal, and response accuracy while faster reaction time over the course of the training sessions. This finding supports the specificity of dACC up-regulation training effects on selective attention in our experimental group. In fact, these effects in the inverse condition control group may reflect disengagement from the MSIT as a result of how reward points were calculated in this group (i.e., based on down-regulation of BOLD dACC activity). Also, these behavioral effects in the inverse condition control group do not support simple practice effects. Moving forward, however, only a neurofeedback-free control group (i.e., a group that receives no neurofeedback during the training sessions or receives sham-neurofeedback based on pre-calculated feedback from, for example, a previous subject) will be necessary to fully exclude practice effects, which would be reflected in greater behavioral improvement in the experimental than the neurofeedback-free control group.

Inconsistent with our prediction, young adults in the up-regulation condition (age control group) did not show neurofeedback training success nor evidence of behavioral benefit. Rather, young adults in the up-regulation condition showed a linear decrease in their reward points within sessions across runs. To interpret this unexpected finding, it is important to note that young participants in the up-regulation condition started at a higher level on each training day than older participants in the up-regulation condition, which could have limited the potential for neurofeedback training success in these young participants over the course of the sessions. Alternatively, it is possible that for young adults, selective attention is not a particularly relevant cognitive facility. In contrast, adults later in life experience significant declines in selective attention, impacting their daily functioning and rendering selective attention a crucial cognitive capacity in aging [2,3]. 

Relatedly, near-ceiling performance on the MSIT in young participants combined with the repetitiveness and length of the MSIT as administered in this project may have resulted in motivational fatigue and neural habituation (i.e., a reduction in neural response to stimuli due to repeated exposure [59]) in young adults, which may have been particularly applicable in the present study given that the dACC is sensitive to novelty and salience [11,60]. In fact, our post hoc analysis of participants’ motivation ratings showed a significant negative linear effect of run on motivation within a given session. In particular, young participants in the up-regulation condition reported increasingly less motivation as the training unfolded, which was not the case for older adults in this training condition (the experimental group). We believe that this finding is particularly interesting and could be reflective of important motivational differences between young and older adults regarding performance in selective attention. To systematically test this possible interpretation that young adults did not show the volitional up-regulation capacity as well as behavioral benefit from the neurofeedback that we had expected based on the literature [31,61], future research could compare neurofeedback success and behavioral benefit across cognitive domains of particular relevance/subjective motivation for young vs. older adults, to specifically increase understanding of factors like motivation in their impact on neurofeedback training success in adults of different ages. Also, future training protocols could benefit from adaptive testing that considers person-specific task engagement, difficulty, and training length. 

Despite its various strengths, the present study also had some limitations, which can be addressed in future research. Our design resulted in a large number of sessions/runs with many repeated trials, going well beyond previous approaches [22] and enhancing statistical power to detect within-subject/participant-level neurofeedback training effects, as a primary goal of our study. However, our sample size of 46 participants, while not particularly small compared with the majority of published papers in the field of rtfMRI neurofeedback training (typically between 11 and 35 [22]; but see Lam et al. [62] for an exception), had limited statistical power to detect between-group/group-level differences. In particular, according to a sensitivity analysis, our sample size resulted in a power of 80% to detect a significant training group by session interaction with a small-to-medium effect size (Cohen’s *f* = 0.18). 

Also, and with possible direct relevance to the unexpected findings of a lack of rtfMRI neurofeedback training success and behavioral benefits in the age control group, the sample size of our experimental group was larger than that of our control groups. This unequal randomization, borrowed from randomized controlled trial designs with large treatment costs [41], allowed us to oversample the experimental group to improve statistical power for the detection of within-subject/participant-level neurofeedback training effects, enhanced information acquisition regarding our primary group of interest, as well as led to significant cost savings. However, at the same time, this unbalanced design limited analytic power in the analysis of our control groups, which could have led to missed effects given relatively smaller samplessizes in these groups, and results have to be interpreted with caution. That is, while we had a power of 80% to detect a small-to-medium effect size in our experimental group (Cohen’s *f* = 0.25), sensitivity analysis showed that we were only able to detect a medium effect size (Cohen’s f > 0.33) in each of our three control groups. Use of a larger sample size and balanced randomization in future research will confirm reliability and enhance generalizability of our findings.

Furthermore, although our older adult sample included a few individuals with PD and some with SCD and a family history of Alzheimer’s disease, the small sub-sample size made an independent analysis of these clinical groups impossible. While results remained stable when considering PD and SCD as covariates in control analyses, a larger number of individuals in each of these clinical categories are needed to reliably determine neurofeedback training success and behavioral benefit on selective attention in these individuals (see also [32]) towards evaluation of rtfMRI neurofeedback training as complementary treatment in neurodegenerative disorders [34].

Finally, going beyond the MSIT, future studies can confirm the present study’s findings using alternative selective attention tasks (e.g., Flanker Task) to demonstrate near-transfer training effects [63]. Also, far-transfer training effects on other cognitive functions subserved by dACC (e.g., decision making, emotion processing) [64] have to be established in future work and will broaden the translational impact of this research. 

## 5. Conclusions

Results from this study support both neurofeedback training success and behavioral benefit on selective attention in older age, providing promising proof-of-concept evidence of functional plasticity in dACC specific to rtfMRI neurofeedback up-regulation training toward enhancing cognitive aging. This work also spurs important new hypotheses to address in future research, such as regarding the role of motivation in neurofeedback training across the adult lifespan.

## Figures and Tables

**Figure 1 brainsci-14-00931-f001:**
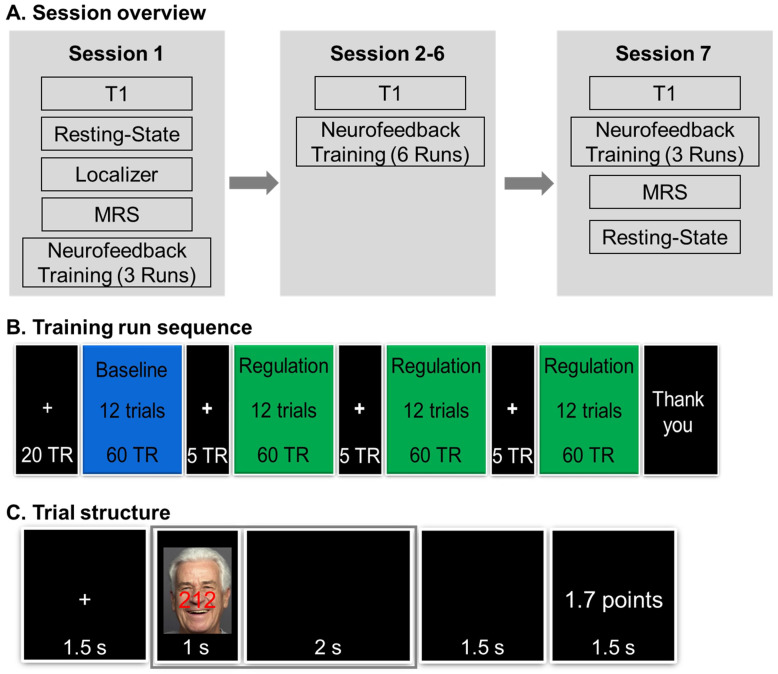
rtfMRI neurofeedback training protocol and task details. (**A**)—*Session Overview*: The neurofeedback training protocol consisted of seven sessions. The T1 scan was 3 min; the resting-state functional scan 8 min; the localizer task comprised two runs, each 8.5 min; the magnetic resonance spectroscopy (MRS) scan approx. 15 min (data from the resting-state and the MRS scans are not reported here). Participants completed 36 training runs in total. (**B**)—*Training Run Sequence*: Each run started with a baseline block, followed by three regulation blocks; each block had 12 trials. (**C**)—*Trial Structure*: Each trial started with a fixation cross presented for 1.5 s, followed by presentation of the trial stimulus for 1 s, followed by a 3.5 s black screen; each trial concluded with the presentation of a feedback screen (showing the reward points) for 1.5 s. The neurofeedback calculation was based on brain data from the 1 s trial stimulus presentation and the following 2 s presentation of a black screen (time window indicated via gray frame in the image).

**Figure 2 brainsci-14-00931-f002:**
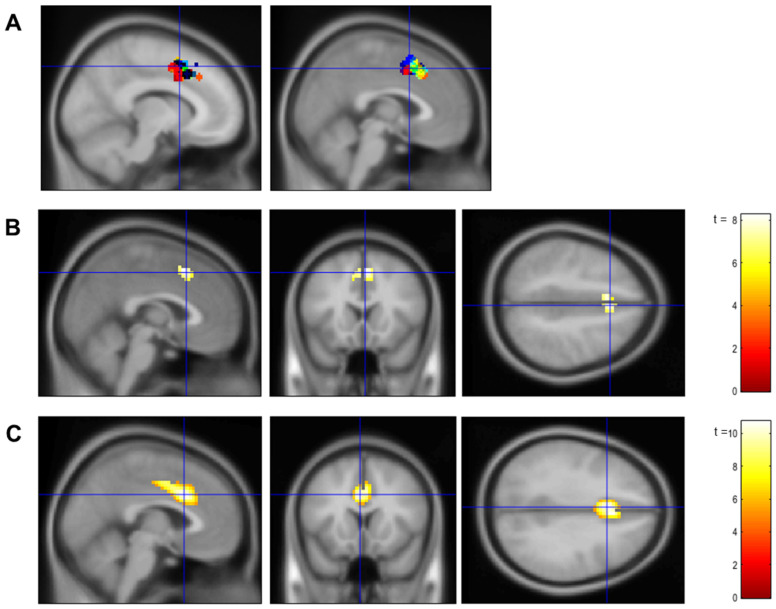
Localization of dACC as target region of interest (ROI) in this neurofeedback training protocol. (**A**): Individualized ROI masks for bilateral dACC from the online analysis of the localizer task; visualized using the MarsBaR toolbox [44]. Individualized ROIs are depicted in different colors. Crosshairs are located at the reference coordinates for left (MNI: x = −10, y = 10, z = 49) and right (MNI: x = 3, y = 10, z = 47) dACC recruited during the MSIT as identified in Bush et al. [8]. (**B**): Cluster of voxels in dACC with greater activity to interference than control trials across all participants (group-level average) derived from the offline GLM analysis of the localizer task. Crosshairs indicate coordinates of peak voxels within the dACC cluster (MNI: x = 2, y = 18, z = 48). (**C**): Cluster of voxels in dACC with greater activity to stimuli (digits/faces) than fixation across all participants during the neurofeedback training runs. Crosshairs indicate coordinates of peak voxels within the dACC cluster (MNI: x = −3, y = 15, z = 33). Activation maps from (**B**,**C**) were derived by using an ACC mask that considered regions of the anterior division of the cingulate gyrus and paracingulate gyrus from the Human Harvard-Oxford atlas [45]. See Appendix A for whole-brain activation maps for the localizer task and the neurofeedback training, respectively. Spectral bars in (**B**,**C**) indicate t-scores.

**Figure 3 brainsci-14-00931-f003:**
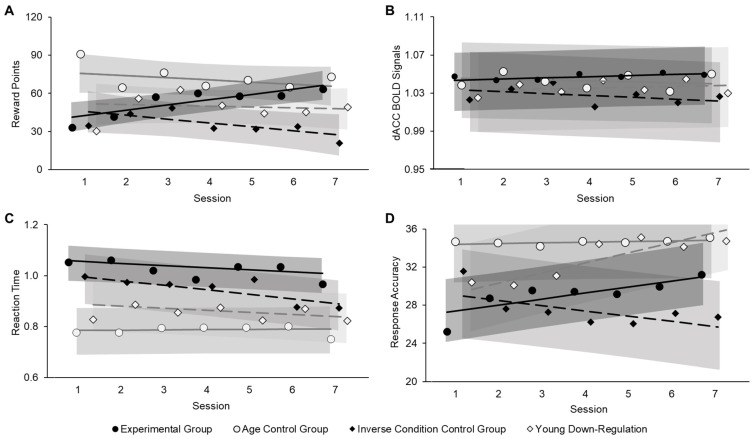
Means and marginal estimates for reward points (**A**), dACC BOLD signal (**B**), reaction time (**C**), and response accuracy (**D**) over the course of the neurofeedback training sessions for the experimental group (black circles and black solid lines), the age control group (white circles and gray solid lines), the inverse condition control group (black diamonds and black dashed lines), and the young down-regulation group (white diamonds and gray dashed lines). Circles and diamonds represent means of measures at each session for each training group. Lines represent marginal estimates of session effects for each training group. Theoretical range of the y-axis was 0 to 360 for reward points (**A**) and 0 to 36 for response accuracy (**D**); note that due to a ceiling effect, error bars for all sessions in the age control group and for sessions 3 to 7 in the young down-regulation group exceeded the theoretical maximum for response accuracy (**D**). Shaded areas indicate 95% confidence intervals.

**Table 1 brainsci-14-00931-t001:** Sample-descriptive information: means (standard deviations) on cognition, affect, and health pre-training in the experimental, age control, inverse condition control, and young down-regulation groups.

	Experimental Group (*N* = 18)	Age Control Group (*N* = 10)	Inverse Condition Control Group (*N* = 9)	Young Down-Regulation Group (*N* = 9)
*Cognition*				
Flanker Inhibitory Control and Attention Test ^a,c^	7.49 (0.95)	8.23 (0.58)	7.97 (0.79)	9.16 (0.63)
Dimensional Change Card Sort Test ^a,b,c^	7.39 (1.08)	8.49 (1.08)	8.44 (1.05)	9.21 (0.65)
List Sorting Working Memory Test ^c^	15.75 (2.21)	16.78 (1.86)	17.43 (2.44)	21.75 (2.55)
Pattern Comparison Processing Speed Test ^a,c^	39.06 (10.48)	48.78 (9.30)	45.57 (6.71)	55.38 (5.40)
*Affect*				
Positive Affect ^c^	3.66 (0.73)	3.14 (0.68)	3.90 (0.89)	2.73 (0.93)
Negative Affect ^c^	1.14 (0.16)	1.24 (0.27)	1.14 (0.16)	1.46 (0.38)
*Health*				
Physical Health	8.00 (1.04)	9.00 (1.20)	8.25 (1.49)	8.44 (1.33)
Mental Health	8.57 (0.85)	8.88 (1.13)	8.50 (2.33)	8.44 (0.88)

Notes. Cognitive measures were assessed via the NIH Cognition Toolbox [42]. Affect were assessed via the Positive and Negative Affect Schedule (PANAS) [43] on a scale from 1 = very slightly or not at all to 5 = extremely. Physical health and mental health were assessed via two single items (i.e., Please rate your general physical health/mental health) on a scale from 1 = poor to 10 = excellent). Superscript a indicates a significant difference between the experimental group and the age control group; superscript b indicates a significant difference between the experimental group and the inverse condition control group; superscript c indicates a significant difference between the experimental group and the young down-regulation group, all at *p* < 0.05.

## Data Availability

The behavioral data and extracted brain data presented in this study are openly available on OSF at https://osf.io/f69wp/?view_only=267f4db77e2c4ce18f04fc42179f63e3, accessed on 10 September 2020, reference number f69wp.

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
