# Peer review of "Real-Time fMRI Neurofeedback Training of Selective Attention in Older Adults"

_brainsci, 2024, doi:10.3390/brainsci14090931_

Round 1

Reviewer 1 Report

Comments and Suggestions for Authors

In this study, the authors aim to show that attention declines with age, due to age-related functional changes in the dorsal anterior cingulate cortex (dACC). This research area is very attractive and promising, and the studies on this topic are worth to support. The overall structure is very well-designed. However, there are some concerns about the paper, as given below:

·        Authors should provide the full name of the MRS where it appears the first time. However, we see it on page 6.

·        The authors should clear the motivation of selection of the happy or angry faces rather than to select the other six naturalistic emotional expressions from the FACES database. Also, please check the direct link on the p6 line 284.

·        Please clarify the down-regulation in detail throughout the manuscript.

·        Please also provide all F contrasts in the supplementary document.

·        It would be better to provide the statistical power analysis for the sample size of the study.

Author Response

  1. Authors should provide the full name of the MRS where it appears the first time. However, we see it on page 6.

The abbreviation MRS is now spelled out as Magnetic Resonance Spectroscopy when first mentioned in the text (p. 6).

  1. The authors should clear the motivation of selection of the happy or angry faces rather than to select the other six naturalistic emotional expressions from the FACES database. Also, please check the direct link on the p6 line 284.

We appreciate this opportunity to clarify our selection of happy and angry faces for use in this study. As now noted on p. 8: “We selected happy and angry facial expressions to align with Ebner and Johnson[52] which previously used this task in young and older adults to determine age-related differences in multisource interference/selective attention”.

We also have updated the link to the FACES database (see p. 8). 

  1. Please clarify the down-regulation in detail throughout the manuscript.

We apologize if the description of the down-regulation training was not clear previously. In the manuscript we now clarify (p. 8): “Identical for both the up- and down-regulation training condition, participants were asked to work as quickly and accurately as possible and to stay focused on the numbers in each trial. Participants were also informed that after the initial trials for each run, trials would award points based on their ability to regulate their brain activity and that the goal was to obtain as many points as possible”.

We then further continue on p. 9: “Note that calculation of the neurofeedback was identical for the up- and the down-regulation condition, other than that the neurofeedback (in the form of reward points) for the up-regulation protocol (i.e., experimental group, age control group) was based on an increase of dACC BOLD signal (relative to BOLD signal in PAC) in regulation relative to baseline blocks; while neurofeedback (in the form of reward points) for the down-regulation protocol (i.e., inverse condition control group, young down-regulation group) was based on a decrease of dACC BOLD signal (relative to BOLD signal in PAC) in regulation relative to baseline blocks”.

  1. Please also provide all F contrasts in the supplementary document.

We would like to clarify that no significant clusters were identified in the F-contrast (at p = 0.05 FEW corrected). We accordingly note on p. 11:No significant clusters at p < 0.05 FWE-corrected were identified for the t-contrast young vs. older, the t-contrast up- vs. down-regulation, or the F-contrast age × training condition at pre-training, supporting successful randomization into our four groups.

  1. It would be better to provide the statistical power analysis for the sample size of the study.

We appreciate this recommendation to elaborate more on power of our analyses.

First, as noted in-text (p. 16), “Our design resulted in a large number of sessions/runs with many repeated trials, going well-beyond previous approaches[22] and enhancing statistical power to detect within-subject/participant-level neurofeedback training effects, as a primary goal of our study.”

In addition, in response to this comment, we have conducted sensitivity analysis and have now reported its result in text (p. 16):In particular, according to a sensitivity analysis, our sample size resulted in a power of 80% to detect a significant training group by session interaction with a small-to-medium effect size (Cohen’s f = 0.18).”

Reviewer 2 Report

Comments and Suggestions for Authors

The study titled "Real-Time fMRI Neurofeedback Training of Selective Attention in Older Adults" investigates the effectiveness of real-time functional magnetic resonance imaging (rtfMRI) neurofeedback in enhancing selective attention in older adults. The study involved 19 young adults and 27 older adults, training them to up- or down-regulate the bilateral dorsal anterior cingulate cortex (dACC) activity during a selective attention task. Results showed that older adults in the up-regulation group experienced significant improvements in selective attention, as evidenced by increased reward points, higher dACC BOLD signals, faster reaction times, and better response accuracy. These improvements were not observed in the down-regulation group or young adults, suggesting the specificity of dACC up-regulation for older adults. The findings support the potential of rtfMRI neurofeedback to enhance cognitive functions in aging, demonstrating functional plasticity in the dACC of older adults.

Suggestions and comments :

Explain the choice of unequal randomization in more detail and discuss its implications for statistical power and generalizability. Including a neurofeedback-free control group would provide a stronger basis for attributing observed effects to the neurofeedback intervention itself.

Provide more context for the statistical analyses, particularly for readers who may not be familiar with multilevel regression models. Including effect sizes alongside p-values would help convey the practical significance of the findings.

Provide complete limitations and future directions statements. 

Author Response

  1. Explain the choice of unequal randomization in more detail and discuss its implications for statistical power and generalizability. Including a neurofeedback-free control group would provide a stronger basis for attributing observed effects to the neurofeedback intervention itself.

We appreciate this opportunity to highlight our choice for unequal randomization and its implications; as well as to reflect on the use of a neurofeedback-free control group in future research. In particular, we note in the Methods on p. 3: Leveraging methodological approaches in clinical randomized controlled trials[42], we employed unequal randomization, resulting in unbalanced group sizes by over-sampling the experimental group (i.e., older adults in the up-regulation condition) at a 2:1 ratio compared to the other/control groups to (i) enhance statistical power for detecting within-subject/participant-level neurofeedback training effects in this primary group of interest; (ii) increase the amount of information on the new neurofeedback training approach in this group; while (iii) allowing for considerable cost savings.

We further note in the Discussion on p. 16: “Also, and with possible direct relevance to the unexpected findings of a lack of rtfMRI neurofeedback training success and behavioral benefits in the age control group, the sample size of our experimental group was larger than that of our control groups. This unequal randomization, borrowed from randomized controlled trial designs with large treatment costs [42], allowed us to oversample the experimental group to improve statistical power for the detection of within-subject/participant-level neurofeedback training effects, enhanced information acquisition regarding our primary group of interest, as well as led to significant cost savings. However, at the same time this unbalanced design limited analytic power in the analysis of our control groups, which could have led to missed effects given relatively smaller samples sizes in these groups; and results have to be interpreted with caution.

We also have now added to the Discussion on p. 16: “Use of a larger sample size and balanced randomization in future research will confirm reliability and enhance generalizability of our findings.

Further, we agree with the reviewer that inclusion of a neurofeedback-free control group would allow us to fully confirm that the observed behavioral effects in the experimental group were indeed a result of the neurofeedback training. That is, while the reported results of our brain-behavioral analysis suggests that behavioral performance improvement shown by the experimental group unlikely reflect practice, our current study design is unable to completely rule out a practice effect. Accordingly, on p. 15 we now write: “As noted, we observed evidence for both neurofeedback training success and behavioral benefit in the experimental group, in that older adults in the up-regulation condition demonstrated a linear increase in both reward points and dACC BOLD signal as well as faster reaction time and better response accuracy over the course of the training sessions. Our brain-behavior analysis, furthermore, showed that greater dACC activation during the neurofeedback training trials was directly correlated with faster and more accurate responding in the MSIT in the experimental group. This pattern of results renders simple practice effects as an explanation of the findings unlikely.

The inverse condition control group, in contrast, showed declines in reward points, dACC BOLD signal, and response accuracy while faster reaction time over the course of the training sessions. This finding supports the specificity of dACC up-regulation training effects on selective attention in our experimental group. In fact, these effects in the inverse condition control group may reflect disengagement from the MSIT as a result of how reward points were calculated in this group (i.e., based on down-regulation of BOLD dACC activity). Also, these behavioral effects in the inverse condition control group do not support simple practice effects. Moving forward, however, only a neurofeedback-free control group (i.e., a group that receives no neurofeedback during the training sessions or receive sham-neurofeedback based on pre-calculated feedback from for example a previous subject) will be necessary to fully exclude practice effects, which would be reflected in greater behavioral improvement in the experimental than the neurofeedback-free control group.

  1. Provide complete limitations and future directions statements. 

As suggested, there is now a section on limitations and future directions, which reads (p. 16-17): “Despite its various strengths, the present study also had some limitations, which can be addressed in future research. Our design resulted in a large number of sessions/runs with many repeated trials, going well-beyond previous approaches[22] and enhancing statistical power to detect within-subject/participant-level neurofeedback training effects, as a primary goal of our study. However, our sample size of 46 participants, while not particularly small compared with the majority of published papers in the field of rtfMRI neurofeedback training (typically between 11-35[22]; but see Lam et al. [63] for an exception), had limited statistical power to detect between-group/group-level differences. In particular, according to a sensitivity analysis, our sample size resulted in a power of 80% to detect a significant training group by session interaction with a small-to-medium effect size (Cohen’s f = 0.18).

Also, and with possible direct relevance to the unexpected findings of a lack of rtfMRI neurofeedback training success and behavioral benefits in the age control group, the sample size of our experimental group was larger than that of our control groups. This unequal randomization, borrowed from randomized controlled trial designs with large treatment costs [42], allowed us to oversample the experimental group to improve statistical power for the detection of within-subject/participant-level neurofeedback training effects, enhanced information acquisition regarding our primary group of interest, as well as led to significant cost savings. However, at the same time this unbalanced design limited analytic power in the analysis of our control groups, which could have led to missed effects given relatively smaller samples sizes in these groups; and results have to be interpreted with caution. Use of a larger sample size and balanced randomization in future research will confirm reliability and enhance generalizability of our findings.

Furthermore, although our older adult sample included a few individuals with PD and some with SCD and a   family history of Alzheimer’s Disease, the small sub-sample size made an independent analysis of these clinical groups impossible. While results remained stable when considering PD and SCD as covariates in control analyses, a larger number of individuals in each of these clinical categories are needed to reliably determine neurofeedback training success and behavioral benefit on selective attention in these individuals (see also [32]), towards evaluation of rtfMRI neurofeedback training as complementary treatment in neurodegenerative disorders[34].

Finally, going beyond the MSIT, future studies can confirm the present study’s findings using alternative selective attention tasks (e.g., Flanker Task) to demonstrate near-transfer training effects[64]. Also, far-transfer training effects on other cognitive functions subserved by dACC (e.g., decision making, emotion processing) [65] have to be established in future work and will broaden translational impact of this research.”

Reviewer 3 Report

Comments and Suggestions for Authors

The article aims to investigate the effectiveness of real-time fMRI neurofeedback training of selective attention in older adults.

The article outlines two objectives of the study. The first is to test the aging brain's capacity to learn volitional up-regulation of dACC activity via rtfMRI neurofeedback training in the context of a selective attention task. The second one was to examine subsequent effects of neurofeedback training on enhancing selective attention in older age. It is noteworthy that under each objective the authors put forward a hypothesis, which is tested in the course of their research.

The topic that is developed in this article is very important and relevant from both fundamental and applied points of view. The article provides new results about an important gap - the effectiveness of rtfMRI neurofeedback training in older adults.

Compared with other published material, this study adds to the subject area the information about the applicability of the concept of rtfMRI neurofeedback training in cognitive aging, supporting both neuofeedback training success and behavioral benefit.

The conclusions are consistent with the evidence and arguments presented.

The references are appropriate. The authors do not provide a separate conclusions section. I recommend that they do. It would also be good to have a separate limitations section.

Author Response

  1. The article aims to investigate the effectiveness of real-time fMRI neurofeedback training of selective attention in older adults.

The article outlines two objectives of the study. The first is to test the aging brain's capacity to learn volitional up-regulation of dACC activity via rtfMRI neurofeedback training in the context of a selective attention task. The second one was to examine subsequent effects of neurofeedback training on enhancing selective attention in older age. It is noteworthy that under each objective the authors put forward a hypothesis, which is tested in the course of their research.

The topic that is developed in this article is very important and relevant from both fundamental and applied points of view. The article provides new results about an important gap - the effectiveness of rtfMRI neurofeedback training in older adults.

Compared with other published material, this study adds to the subject area the information about the applicability of the concept of rtfMRI neurofeedback training in cognitive aging, supporting both neuofeedback training success and behavioral benefit. The conclusions are consistent with the evidence and arguments presented.

The references are appropriate.

We very much appreciate the reviewer’s positive evaluation of our paper and the time they have invested in reviewing it.

    2. The authors do not provide a separate conclusions section. I recommend that they do. It would also be good to have a separate limitations section.

As suggested, there are now separate sections for both study limitations and conclusions. Those sections read:

Limitations (p. 16-17): “Despite its various strengths, the present study also had some limitations, which can be addressed in future research. Our design resulted in a large number of sessions/runs with many repeated trials, going well-beyond previous approaches[22] and enhancing statistical power to detect within-subject/participant-level neurofeedback training effects, as a primary goal of our study. However, our sample size of 46 participants, while not particularly small compared with the majority of published papers in the field of rtfMRI neurofeedback training (typically between 11-35[22]; but see Lam et al. [63] for an exception), had limited statistical power to detect between-group/group-level differences. In particular, according to a sensitivity analysis, our sample size resulted in a power of 80% to detect a significant training group by session interaction with a small-to-medium effect size (Cohen’s f = 0.18).

Also, and with possible direct relevance to the unexpected findings of a lack of rtfMRI neurofeedback training success and behavioral benefits in the age control group, the sample size of our experimental group was larger than that of our control groups. This unequal randomization, borrowed from randomized controlled trial designs with large treatment costs [42], allowed us to oversample the experimental group to improve statistical power for the detection of within-subject/participant-level neurofeedback training effects, enhanced information acquisition regarding our primary group of interest, as well as led to significant cost savings. However, at the same time this unbalanced design limited analytic power in the analysis of our control groups, which could have led to missed effects given relatively smaller samples sizes in these groups; and results have to be interpreted with caution. Use of a larger sample size and balanced randomization in future research will confirm reliability and enhance generalizability of our findings.

Furthermore, although our older adult sample included a few individuals with PD and some with SCD and a   family history of Alzheimer’s Disease, the small sub-sample size made an independent analysis of these clinical groups impossible. While results remained stable when considering PD and SCD as covariates in control analyses, a larger number of individuals in each of these clinical categories are needed to reliably determine neurofeedback training success and behavioral benefit on selective attention in these individuals (see also [32]), towards evaluation of rtfMRI neurofeedback training as complementary treatment in neurodegenerative disorders[34].

Finally, going beyond the MSIT, future studies can confirm the present study’s findings using alternative selective attention tasks (e.g., Flanker Task) to demonstrate near-transfer training effects[64]. Also, far-transfer training effects on other cognitive functions subserved by dACC (e.g., decision making, emotion processing) [65] have to be established in future work and will broaden translational impact of this research.”

Conclusions (p. 17): “Results from this study support both neurofeedback training success and behavioral benefit on selective attention in older age, providing promising proof-of-concept evidence of functional plasticity in dACC specific to rtfMRI neurofeedback up-regulation training toward enhancing cognitive aging. This work also spurs important new hypotheses to address in future research such as regarding the role motivation in neurofeedback training across the adult lifespan.”

Reviewer 4 Report

Comments and Suggestions for Authors

Real-Time fMRI Neurofeedback Training of Selective Attention in Older Adults

 I have read the manuscript with interest, and the authors can find my suggestions, recommendations, and concerns, section by section, as follows:

Introduction: I advise to remove the first sentence since it is quite obvious. Moreover, you introduce the intriguing and interesting role played by the cingulate cortex in the processes and mechanisms related to healthy aging. However, dACC is a complex brain structure, and a wider description is needed. Indeed, the references added are a bit obsolete. For this reason, this paragraph needs to be improved and at least rewritten with more new references and highlighting the role played by this region and the interplay with the other cingulate subregions. The description made by the authors about the use and application of rtfMRI is well-written and interesting. The hypotheses are clear.

In the methods the sample the older adults need to follow a more clear definition. Indeed, you recruited participants in a range 57-86 years. The definition of older adults is >65  years. https://www.nih.gov/nih-style-guide/age#:~:text=The%20National%20Institute%20on%20Aging,these%20terms%2C%20ask%20for%20specifics.

However, as specified you can find different definitions of “older adult”. This needs to be revised and clarified in the text. I suggest at least, to remove one or more outliers from the data analysis and rerun the analysis.

Despite this, the procedure was described in a good way and rigorously. You indeed provided also MRI sequence details that can allow completely the replicability of the study.   

Results: the results are interesting. However, I suggest adding a table summarizing the results and describing them in the main text. This facilitates the reading.

Discussion: in the discussion, the older adults are “late” older adults. Please, refer to the comments about the methods section. The advice is that you homogenize the nomenclature and check this definition.

Moreover, learning during aging, you need to add more information about. Volitional learning in elderly people has been studied from a psychological point of view. Similarly, most of the discussion is based on attention. Attention is processed also in dACC, but this region is also important for decision-making and problem-solving. In older adults, these cognitive functions have some peculiarities that could be highlighted and a bit discussed.   

Author Response

We greatly appreciate the constructive feedback provided by the reviewer and have addressed each comment as detailed below.

  1. Introduction: I advise to remove the first sentence since it is quite obvious.

As recommended, we have removed this first sentence.

2. Moreover, you introduce the intriguing and interesting role played by the cingulate cortex in the processes and mechanisms related to healthy aging. However, dACC is a complex brain structure, and a wider description is needed. Indeed, the references added are a bit obsolete. For this reason, this paragraph needs to be improved and at least rewritten with more new references and highlighting the role played by this region and the interplay with the other cingulate subregions.

Following this suggestion, we have revised the first paragraph in the introduction with updated references, speaking to the role of dACC in other cognitive functions (e.g., executive function, decision making) and its connection with other brain regions/networks (e.g., insula, amygdala, medial prefrontal cortex). This paragraph now reads (p. 1): “It is crucial that individuals allocate their limited attention selectively to relevant information (i.e., targets) and inhibit irrelevant information (i.e., distractors). For example, when having a conversation in a crowded space, we pay attention to what our conversation partner says while blocking out background noise[1]. The capacity for selective attention, however, declines with age[2], [3]. This age-related decline has been attributed to functional changes in the dorsal anterior cingulate cortex (dACC)[4], [5]. Specifically, neuroimaging evidence supports dACC involvement in selective attention tasks, e.g., the Stroop Task[6], the Eriksen Flanker Task[7], and the Multi-Source Interference Task (MSIT)[8], including in older adults[9]. These tasks require cognitive processing such as target detection, novelty/salience indication, and conflict monitoring[10], [11]. In addition, dACC is involved in higher-order complex processes pertaining to executive function, decision making, and cognitive control[12]–[15]; and in these roles has been shown to be functionally connected to other brain regions, such as the insula, amygdala, orbital frontal cortex, medial prefrontal cortex, superior temporal cortex, and thalamus[16] and serving as key node in large-scale brain networks (e.g., the salience network and the default mode network)[17], [18].

3. The description made by the authors about the use and application of rtfMRI is well-written and interesting. The hypotheses are clear.

We were happy to read this. Thank you for this positive feedback!

4. In the methods the sample the older adults need to follow a more clear definition. Indeed, you recruited participants in a range 57-86 years. The definition of older adults is >65 years. https://www.nih.gov/nih-style-guide/age#:~:text=The%20National%20Institute%20on%20Aging,these%20terms%2C%20ask%20for%20specifics. However, as specified you can find different definitions of “older adult”. This needs to be revised and clarified in the text. I suggest at least, to remove one or more outliers from the data analysis and rerun the analysis.

In response to this comment, we have added a footnote (# 1, p. 3) that now clarifies: “According to the National Institutes of Health, older adults are defined as 65 years and older (https://www.nih.gov/nih-style-guide/age#older-adults). In our sample, age ranged from 57-86, with 8 participants were younger than 65. However, for better readability, we refer to our sample of middle-aged and older adults as ‘older adults’ throughout the paper.” 

In addition, while our project was not designed, and statistically powered, to dichotomize middle-age and older adults into two separate groups, we conducted two control analyses pertaining to age. This information can be found in Footnote 4 (p. 13), which reads: “Given the wide age range among older adults in our sample (57-86 years), we conducted two control analyses: (1) one in which we limited the older adult sample to just older adults following the definitions by the NIH (65 years and older; n = 19); and (2) one in which we included chronological age (57-86 years) as covariate. Results from these additional analyses closely aligned with those reported in the main text for all outcome variables.”

5. Despite this, the procedure was described in a good way and rigorously. You indeed provided also MRI sequence details that can allow completely the replicability of the study.   

Thank you! We were excited to read this positive feedback.

6. Results: the results are interesting. However, I suggest adding a table summarizing the results and describing them in the main text. This facilitates the reading.

This is a great suggestion, and we have added Table S4 (see also below) to the Supplementary Materials. This table summarizes all effects from the four models we conducted; two for testing neurofeedback training success (on reward points and dACC BOLD signal, respectively) and two for testing behavioral benefit (on reaction times and response accuracy, respectively).

Table S4. Parameter estimates (B/σ2 (SE), Standard Error [SE]) of effects for reward points, dACC BOLD signal, reaction time, and response accuracy.

Reward  Points

dACC BOLD Signal

Reaction  Time

Response Accuracy

Fixed Effect

B (SE)

B (SE)

B (SE)

B (SE)

Experimental vs. Age Control

16.91 (7.94)

-0.0042 (0.026)

-0.246 (0.059)

5.49 (2.65)

Experimental vs. InverseCondition Control

-17.47 (8.22)

-0.0197 (0.0269)

-0.092 (0.061)

-1.78 (2.74)

Experimental vs. Young Down-Regulation

-4.18 (8.22)

-0.0115 (0.0269)

-0.172 (0.061)

3.65 (2.74)

Session

4.14 (1.13)

0.0012 (0.0005)

-0.008 (0.003)

0.62 (0.13)

Session: Experimental vs. Age Control

-5.83 (1.89)

-0.0021 (0.0009)

0.009 (0.004)

-0.55 (0.22)

Session: Experimental vs. Inverse Condition Control

-7.17 (1.96)

-0.003 (0.0009)

-0.009 (0.005)

-1.15 (0.23)

Session: Experimental vs. Young Down-Regulation

-4.77 (1.96)

-0.0003 (0.0009)

0.0001 (0.005)

0.42 (0.23)

Run

-0.7 (1.26)

0.0001 (0.0006)

0.002 (0.003)

0.29 (0.15)

Run: Experimental vs. Age Control

-9.51 (2.11)

0.0011 (0.001)

0.003 (0.005)

-0.46 (0.25)

Run: Experimental vs. Inverse Condition Control

2.19 (2.19)

-0.0019 (0.001)

0.001 (0.005)

-0.43 (0.25)

Run: Experimental vs. Young Down-Regulation

2.79 (2.19)

0.0011 (0.001)

-0.0001 (0.005)

-0.27 (0.25)

Session × Run

0.28 (0.83)

-0.0003 (0.0004)

-0.002 (0.002)

-0.21 (0.10)

Session × Run: Experimental vs. Age Control

1.48 (1.39)

0.0001 (0.0007)

0.004 (0.003)

0.21 (0.16)

Session × Run: Experimental vs. Inverse Condition Control

1.8 (1.44)

0.0006 (0.0007)

0.003 (0.003)

0.42 (0.17)

Session × Run: Experimental vs. Young Down-Regulation

-1.57 (1.44)

0.0003 (0.0007)

0.00005 (0.003)

0.13 (0.17)

Intercept

53.82 (4.74)

1.047 (0.0158)

1.034 (0.035)

29.11 (1.58)

Random Effect

σ2 (SE)

σ2 (SE)

σ2 (SE)

σ2 (SE)

Intercept

332.28 (84.54)

0.0042 (0.0009)

0.022 (0.005)

44.08 (9.4)

Note. We examined effects of session and run, as well as their interaction, on reward points, dACC BOLD signal, reaction time, and response accuracy via four separate models, with training group as moderator. The experimental group served as reference category, and therefore effects of session and run, as well as their interaction, reflect those in the experimental group. Bold print indicates significant effects at p < 0.05.

Also, a paragraph at the start of the Discussion summarizes the key findings of this study and their implications (p. 14): “Results provide crucial proof of concept of rtfMRI neurofeedback training in cognitive aging, supporting both neurofeedback training success (i.e., higher reward points and corresponding dACC BOLD signal over the course of the training sessions) and behavioral benefit (i.e., faster reaction time and better response accuracy in the MSIT) among older adults trained to up-regulate dACC activity; in a sample with a wider and older age range than most prior work. We further demonstrate specificity of volitional dACC up-regulation training in older adults, as neither older adults in the down-regulation condition (inverse condition control group), nor -- unexpectedly and possibly related to reduced motivation in this group to continue with the neurofeedback training pertaining to selective attention -- young adults in the up-regulation condition (age control group) showed comparable effects.

7. Discussion: in the discussion, the older adults are “late” older adults. Please, refer to the comments about the methods section. The advice is that you homogenize the nomenclature and check this definition.

We apologize for this oversight. The word “late” was not supposed to be there, and we have removed it from the text.

8. Moreover, learning during aging, you need to add more information about. Volitional learning in elderly people has been studied from a psychological point of view. Similarly, most of the discussion is based on attention. Attention is processed also in dACC, but this region is also important for decision-making and problem-solving. In older adults, these cognitive functions have some peculiarities that could be highlighted and a bit discussed.  

This is a great comment! We agree that the implications of neurofeedback training success among older adults as they pertaining to learning in aging is an important aspect of our work and discuss this now as follows (p. 14): “Our instructions to participants entailed that the neurofeedback was based on their real-time brain activity. No explicit instructions were given as to what strategies should be used to drive this activity. Thus, neurofeedback training success observed in the experimental group likely reflects successful implicit learning[25], [55] in these older adults. This finding is in line with literature supporting intact implicit (relative to explicit) learning in aging[56], [57]. It will be interesting to determine in future research, involvement of other brain regions than dACC in this learning process such as the striatum and medial temporal regions[58], [59].

Round 2

Reviewer 1 Report

Comments and Suggestions for Authors

The authors have solved most of the issues in the revised version of the study. Thanks to the authors for collaborating to upgrade the paper. The remaining minor issues are as follows:

·         The direct link the authors provided on p.8 l.277 still does not work when clicking on it. While it seems to be true, however, the full hyperlink address contains “….for details” by mistake (https://faces.mpdl.mpg.de/imeji/%20for%20details).

·         Thanks to the authors for providing the power analysis for their dataset. As far as I understood, they performed the statistical analysis on the whole subject set, i.e. for 46 participants, however, it is better to perform for each group. The authors can benefit from the practical guide studies such as https://doi.org/10.22531/muglajsci.1282492

Author Response

1. The direct link the authors provided on p.8 l.277 still does not work when clicking on it. While it seems to be true, however, the full hyperlink address contains “….for details” by mistake (https://faces.mpdl.mpg.de/imeji/%20for%20details).

Thank you for this feedback. The link is functional now.

2. Thanks to the authors for providing the power analysis for their dataset. As far as I understood, they performed the statistical analysis on the whole subject set, i.e. for 46 participants, however, it is better to perform for each group. The authors can benefit from the practical guide studies such as https://doi.org/10.22531/muglajsci.1282492

In line with this suggestion, we have added on p. 18: “That is, while we had a power of 80% to detect a small-to-medium effect size in our experimental group (Cohen’s f = 0.25), sensitivity analysis showed that at a power of 80% we were only able to detect a medium effect size (Cohen’s f > 0.33) in each of our three control groups”.

Reviewer 4 Report

Comments and Suggestions for Authors

The authors addressed all my concerns 

Author Response

The authors addressed all my concerns 

Thank you again for your constructive feedback and time invested in the review of our paper. It is greatly appreciated.